# A novel trajectory learning method for robotic arms based on Gaussian Mixture Model and k-value selection algorithm

**Jingnan Yan[1], Yue Wu[1]\*, Kexin Ji[1], Cheng Cheng[2], Yili Zheng[1]**

1 School of Technology, State Key Laboratory of Efficient Production of Forest Resources, Key Laboratory of National Forestry and Grassland Administration on Forestry Equipment and Automation, Beijing Forestry University, Beijing, China, 2 School of Artificial Intelligence and Automation, Huazhong University of Science and Technology, Wuhan, China

\* wuyue_a@bjfu.edu.cn

## Abstract

In the field of robotic arm trajectory imitation learning, Gaussian Mixture Models are widely used for their ability to capture the characteristics of complex trajectories. However, one major challenge in utilizing these models lies in the initialization process, particularly in determining the number of Gaussian kernels, or the k-value. The choice of the k-value significantly impacts the model's performance, and traditional methods, such as random selection or selection based on empirical knowledge, often lead to suboptimal outcomes. To address this challenge, this paper proposes a novel trajectory learning method for robotic arms that combines Gaussian Mixture Model with a k-value selection algorithm. The proposed approach leverages the principles of the elbow method along with the properties of exponential functions, correction terms, and weight adjustments to accurately determine the optimal k-value. Next, k-means clustering is applied with the optimal k-value to initialize the parameters of the Gaussian Mixture Model, which are then refined and trained through the Expectation-Maximization algorithm. The resulting model parameters are then employed in Gaussian Mixture Regression to generate the robotic arm trajectories. The effectiveness of the proposed method is validated through both simulation experiments with two-dimensional theoretical nonlinear dynamic systems and physical experiments with actual robotic arm data. Experimental results demonstrate that, compared to the traditional Gaussian Mixture Model approach, the proposed method improves trajectory accuracy by more than 15%, as shown by reductions in both the Mean Absolute Error and the Root Mean Square Error. These results highlight that the proposed method significantly enhances the accuracy and efficiency of robotic arm trajectory generation, providing a promising solution for improving robotic manipulation tasks.

## 1. Introduction

With the rapid development of robotics technology, robotic arms are increasingly being used in fields such as industry, agriculture, healthcare, and services [1,2]. Enabling robotic arms to autonomously learn and imitate complex human motion trajectories is a significant topic in

**Data availability statement:** All relevant data are within the manuscript and its Supporting Information files.

**Funding:** This work was supported by the Fundamental Research Funds for the Central Universities of China (BLX202128), and the National Natural Science Foundation of China under Grant 62273053.

robotics. By introducing trajectory imitation learning methods, robotic arms can learn complex motion patterns from human demonstrations and adjust and optimize them according to specific task requirements [3]. This allows robotic arms to exhibit higher adaptability in constantly changing environments, thereby improving the execution efficiency of complex tasks. Therefore, in-depth research and development of trajectory imitation learning methods are of great importance not only for enhancing the intelligence of robotic arms but also for providing technical support and assurance for their widespread application across various fields.

In early research [4–6], trajectory imitation learning primarily relied on simple interpolation and polynomial fitting methods. Su [7] developed a manipulator trajectory planning technique using algebraic-trigonometric Hermite polynomials, demonstrating that this method ensures Ck-continuity and enables the generation of smooth and continuous manipulator movements through curve-based data point interpolation. Walambe [8] developed a spline-based trajectory generation method for a car-type mobile robot, overcoming parametric singularities and providing a smooth and optimized motion planning approach. Although these methods achieved certain successes in early studies, they often performed poorly when dealing with high-dimensional data and complex trajectories [9].

To overcome these limitations, researchers have started exploring more advanced methods. Among them, the Gaussian Mixture Model (GMM) has gradually become an important method in trajectory imitation learning due to its powerful trajectory encoding capability, which can effectively handle multi-modal data and capture complex trajectory features [10–14]. Zhang [15] proposed a unified framework for learning complex tasks, which first uses a Bayesian inference-based change point detection algorithm to segment unstructured demonstrations into motion primitives online, and then models them using GMM. Yang [16] introduced an enhanced robotic skill learning system combining Dynamic Movement Primitive (DMP), GMM, and Gaussian Mixture Regression (GMR), and designed a controller based on Radial Basis Function Neural Networks (RBFNN) to improve trajectory tracking accuracy. Kyrarini [17] developed a robot learning framework leveraging Gaussian Mixture Models for real-time adaptation to human-robot collaboration in industrial assembly tasks, allowing for dynamic response to changes in the environment and object manipulation without the need of additional training by demonstration. Cheng [18], aiming to solve the problem of online adaptive path planning in dynamic environments, proposed a path planning method that generates smooth paths using GMM and GMR, and employs an improved Probabilistic Roadmap (PRM) for local path planning during the online phase. Su [19] developed a novel methodology for teaching robotic systems surgical skills through multiple human demonstrations, leveraging Gaussian Mixture Models to model 3-D manipulation and ensuring compliance with the remote center of motion constraint in robot-assisted minimally invasive surgery. Bai [20] presented an automatic valve rotation control strategy for robots based on teaching-learning, encompassing three stages: teaching, model learning, and task repetition. This strategy aligns teaching data using dynamic time warping, and utilizes DMP combined with GMM and GMR for data analysis. Jia [21], addressing the challenge of robot arm trajectory planning in unstructured environments, proposed a method that utilizes the Gaussian Mixture Model and Gaussian Mixture Regression to generate ideal primitive trajectory actions, significantly improving the adaptability and generalization performance of robotic arms. When applying GMM, initializing with k-means clustering is a common step [22], which can effectively determine the initial positions of Gaussian kernels, thereby improving the accuracy of the final trajectory learning results. However, the number of Gaussian kernels needs to be predetermined, and an inappropriate selection can affect the final generated trajectory.

In order to achieve more accurate trajectory learning, this paper proposes a novel robotic arm trajectory learning method GMM-KVS that combines Gaussian mixture model and

k-value selection algorithm. The selection of the k-value, which determines the number of Gaussian kernels, is a critical step in GMM-based trajectory learning. Traditional methods, such as random selection or empirical estimation, often lead to suboptimal results, negatively impacting trajectory accuracy. The k-value selection algorithm introduced in this paper optimizes the number of Gaussian kernels, ensuring more accurate model initialization and improved trajectory learning. This advancement is particularly beneficial for real-world applications like industrial assembly lines and autonomous logistics robots, where precise and adaptable trajectories are crucial. The proposed method enhances trajectory accuracy and model robustness, significantly improving robotic arm performance in dynamic environments. The main contributions of this paper can be summarized as follows.

1. In the trajectory learning method for robotic arms based on Gaussian mixture models, a k-value selection algorithm is used to determine the optimal number of Gaussian kernels, thereby improving the accuracy of the final trajectory learning results.

2. In order to overcome the situation where the elbow method may not be able to determine the optimal k-value when applied to specific datasets, the exponential function characteristics, correction terms, and weight adjustment are used to improve it and enhance the accuracy of selecting the optimal k-value.

3. The effectiveness of the proposed method was verified through simulation and physical experiments.

The rest of this paper is organized as follows. Section 2 outlines the problem statement and preliminaries. Section 3 introduces the proposed GMM-KVS method. Section 4 conducted simulations and physical experiments, and presented the experimental results. Finally, Section 5 concludes the work.

## 2. Preliminaries and problem statement

Robotic arm trajectory learning plays a crucial role in robotics. By imitating and learning human-demonstrated motion trajectories, robotic arms can autonomously perform various complex tasks. The Gaussian Mixture Model (GMM), as an effective probabilistic model, is widely used in trajectory learning tasks. By modeling the probability distribution of the demonstrated trajectories, GMM enables precise learning and generation of robotic arm trajectories, enhancing the performance of robotic arms in practical tasks.

The Gaussian Mixture Model is a probabilistic density estimation model suitable for multi-modal data [23]. GMM assumes that the data is composed of multiple Gaussian distributions, with each Gaussian distribution describing a subset of the data. The probability density function of GMM is expressed as

$$p(x) = \sum_{k=1}^{K} \pi_k \phi(x \mid \mu_k, \Sigma_k) \tag{1}$$

where $\pi_k$ is the weight of the k-th Gaussian distribution, $\mu_k$ and $\Sigma_k$ are the mean and covariance matrix, respectively, and K is the number of Gaussian distributions.

In GMM-based trajectory learning methods, model initialization is one of the key steps. Using k-means clustering method to preliminarily cluster the original trajectory data can determine the initial parameters of GMM. K-means clustering is a widely used unsupervised learning algorithm [24] primarily used to partition a dataset into k distinct clusters, each represented by its mean (centroid). The algorithm iteratively optimizes the clustering so that

data points within the same cluster are as similar as possible, while data points from different clusters are as dissimilar as possible. The main steps are as follows:

1) Select k initial centroids: Randomly choose k data points as the initial centroids.

2) Assign data points: Assign each data point to the cluster of the nearest centroid.

3) Update centroids: Recalculate the centroid of each cluster, which is the mean of all data points belonging to that cluster.

4) Iterate: Repeat steps 2 and 3 until the centroids no longer change significantly or a pre-defined number of iterations is reached.

Upon completion of the iterations, the centroid of each cluster serves as the mean vector of the corresponding Gaussian distribution. The covariance matrix is calculated based on the distribution of data points within each cluster, and the ratio of the number of data points in each cluster to the total number of data points is used as the mixing coefficient.

When applying k-means clustering, the number of initial centroids, or Gaussian kernels, is typically predefined. The number of Gaussian kernels (k-value) can significantly impact the accuracy of both the GMM and trajectory learning. Therefore, determining the optimal k-value $K_{best}$ for the GMM is crucial before proceeding with the subsequent steps of robotic arm trajectory learning. An appropriate method must be employed to obtain $K_{best}$, ensuring the accuracy of the final generated trajectory. Based on this, a novel robotic arm trajectory learning method, GMM-KVS, is proposed by combining the Gaussian Mixture Model with a k-value selection algorithm. This method enables the optimal selection of the k-value and the generation of the final trajectory through learning from the original trajectory data.

## 3. The proposed GMM-KVS method

The flowchart of the GMM-KVS method for robotic arm trajectory imitation learning is shown in Fig 1. The demonstrator first drags the robotic arm to collect the demonstrated trajectory data. Then, the optimal k-value is obtained using the k-value selection algorithm,

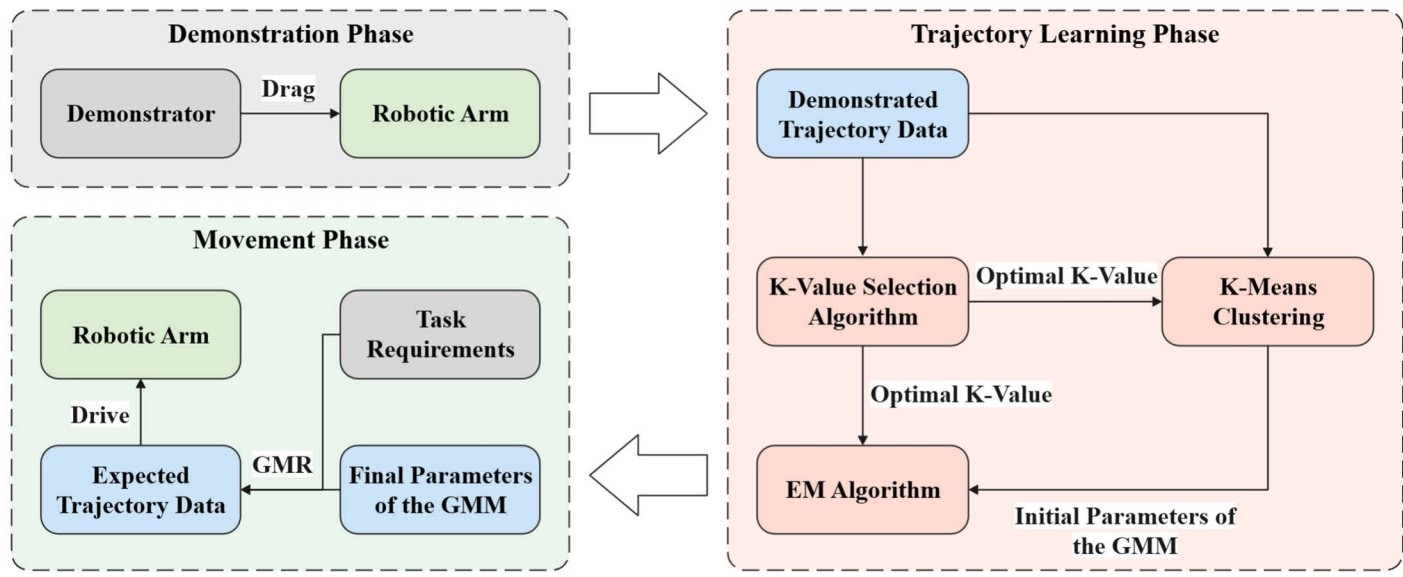

**Fig 1. GMM-KVS method flowchart.**

followed by initializing with k-means clustering. The initialized GMM parameters are then trained using the Expectation-Maximization (EM) algorithm to obtain the final parameters. Depending on the specific task requirements, Gaussian Mixture Regression (GMR) is used to obtain the expected trajectory, ultimately enabling the robotic arm to complete the task.

### 3.1. K-value selection and k-means clustering

The elbow method is a k-value selection algorithm for k-means clustering based on the Sum of Squared Errors (SSE) [25]. Specifically, when using the elbow method, different k-values are first input into the k-means clustering algorithm to obtain the corresponding clustering results. Then, the SSE for the clustering results of each k-value is calculated, and a line graph is plotted showing the relationship between the k-values and their corresponding SSE.

When k is less than the optimal number of clusters, increasing the k-value significantly improves clustering compactness, resulting in a sharp decrease in SSE. When k is greater than the optimal number of clusters, increasing the k-value has a diminishing effect on improving clustering compactness, and the SSE decrease rate slows down noticeably. Therefore, the relationship between k and SSE forms an elbow-shaped line graph, as shown in Fig 2. The k-value at the "elbow point" corresponds to the optimal k-value.

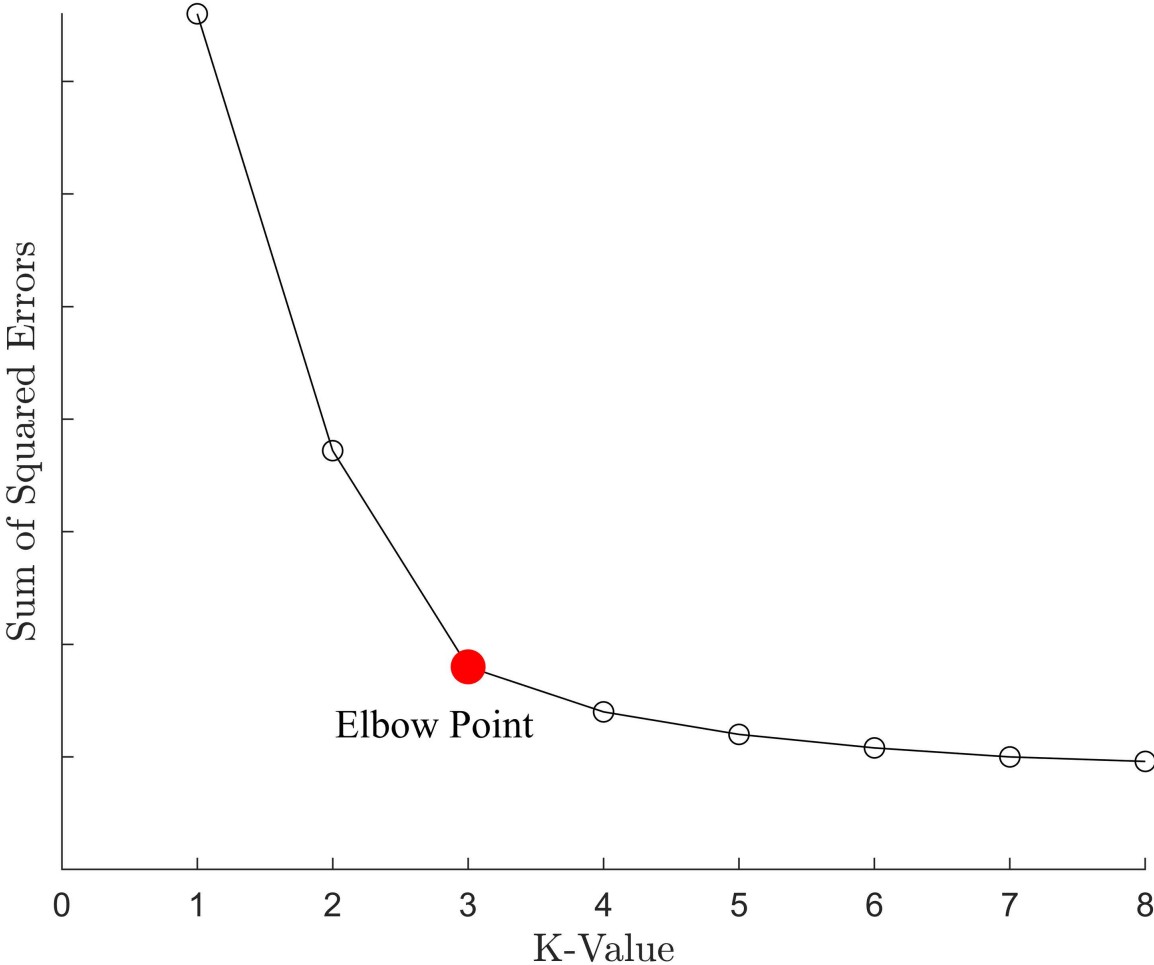

**Fig 2. Schematic diagram of the elbow method.**

When using the elbow method, the distance from each sample object in each group to the centroid of its respective group is first calculated. Then, these distances are squared and summed to obtain the SSE for each group. The purpose is to measure the compactness of each group. The SSE formula for the i-th group is as follows:

$$SSE_i = \sum_{p \in A_i}^{n} \left( p - a_i \right)^2 \tag{2}$$

where p is a sample object in group $A_i$, and $a_i$ is the centroid of the current group. The total Sum of Squared Errors for K groups can then be calculated as follows:

$$SSE = \sum_{i=1}^{K} SSE_i \tag{3}$$

By calculating the SSE for different k-values and plotting the results, the optimal k-value can be determined. Although the traditional elbow method is computationally efficient, it may exhibit unclear "elbow point" for certain specific datasets, as shown in Fig 3. In such cases, selecting the k-value may result in significant deviations, thereby affecting the final results.

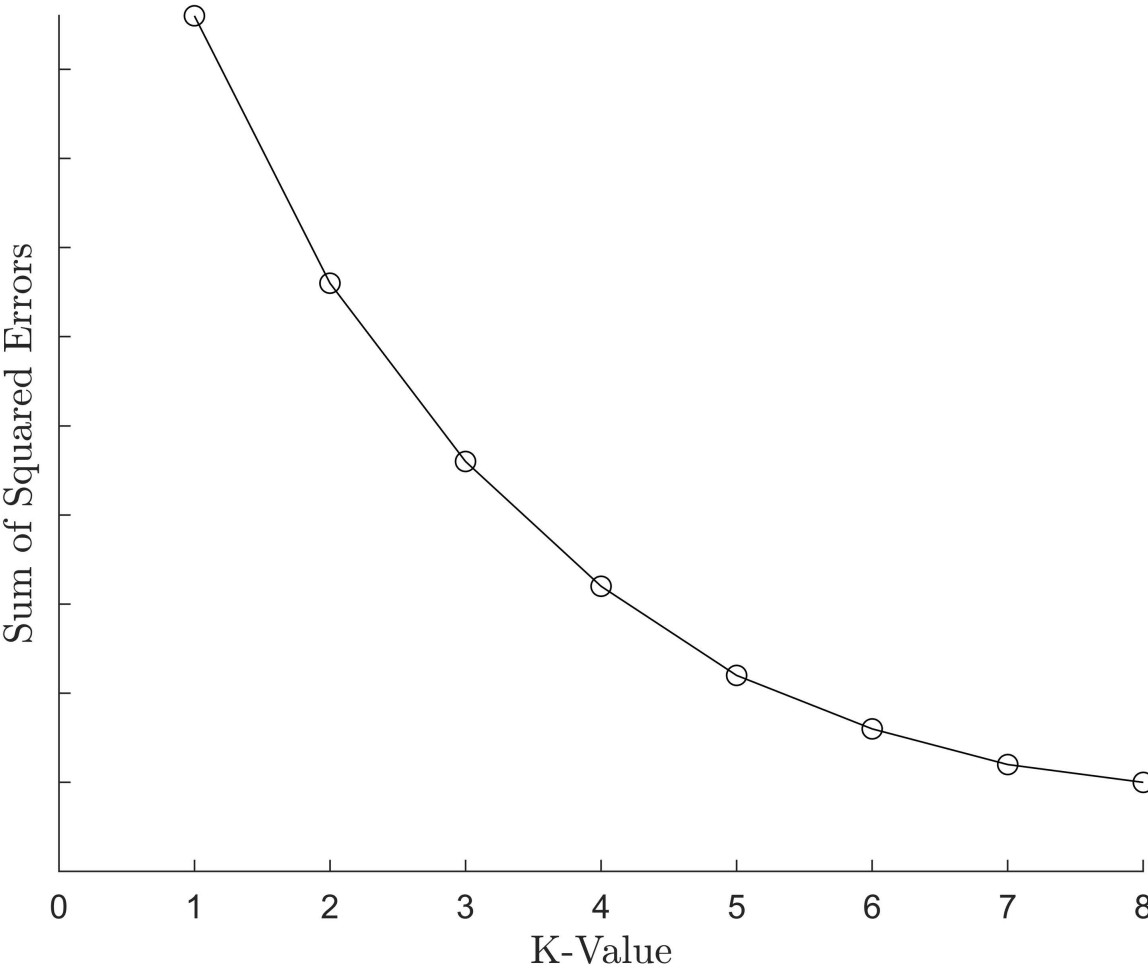

**Fig 3. Schematic diagram of the elbow method (when the effect is poor).**

The traditional elbow method's evaluation criterion is based on the total Sum of Squared Errors, which sums the within-cluster SSEs of all groups. If, with the same number of clusters, grouping A has one cluster with a large $SSE_i$ and another with a small $SSE_j$, while grouping B has clusters with similar $SSE_i$ values, and the total SSEs of both groupings are similar, grouping B is generally considered better. However, SSE alone cannot make this distinction. Therefore, the traditional elbow method should be improved to increase its general applicability and maintain good performance under such conditions.

Considering that the exponential function $e^x$ is highly sensitive to positive changes in the exponent, the exponential function is introduced to modify the calculation process [26]. By applying $e^x$ to the SSE calculation formula, the influence of poorly clustered groups is amplified, thereby giving more weight to these groups during the k-value selection. The modified formula is as follows:

$$M_{SSE} = \sum_{i=1}^{K} e^{\sum_{p \in A_i}^{n} |p - a_i|^2} \tag{4}$$

To further adjust the influence of clusters that exhibit poor compactness, a correction term is introduced. This term increases the weight of clusters with higher SSE values by using the maximum SSE among the K groups as the exponent's input for the exponential function. The correction term formula is as follows:

$$P_{SSE} = e^{\max_{1 \le i \le K} \sum_{p \in A_i}^{n} |p - a_i|^2} \tag{5}$$

Combining (4) and (5), the improved formula is obtained, which incorporates both the exponential adjustment and the correction term to better reflect the impact of poorly clustered groups on the overall SSE. The resulting formula is as follows:

$$MP = \sum_{i=1}^{K} e^{\sum_{p \in A_i}^{n} |p - a_i|^2} + e^{\max_{1 \le i \le K} \sum_{p \in A_i}^{n} |p - a_i|^2} \tag{6}$$

where p is a sample object in group $A_i$, $a_i$ is the centroid of the current group, K is the number of clusters, $\sum_{p \in A_i}^{n} |p - a_i|^2$ is the SSE of the current group, and max denotes the maximum SSE among all groups.

Considering that in practical situations, some points may be far from their group's centroid, the introduction of the exponential function could cause an "exponential explosion" phenomenon, which may disproportionately affect the k-value selection and the subsequent trajectory learning process. To address this potential issue, a scaling weight $\theta$ is introduced in (6) to moderate the influence of the exponential term. The scaling weight $\theta$ serves to reduce the impact of extreme distances, preventing overly large values from distorting the final calculation. This adjustment ensures a more balanced and controlled selection of the optimal k-value, enhancing the stability and robustness of the method. The final k-value selection algorithm formula is as follows:

$$S_{MP} = \sum_{i=1}^{K} e^{\frac{\sum_{p \in A_i}^{n} |p - a_i|^2}{\theta}} + e^{\frac{\max_{1 \le i \le K} \sum_{p \in A_i}^{n} |p - a_i|^2}{\theta}} \tag{7}$$

After selecting the optimal k-value, k-means clustering is used to obtain the initial parameters for the GMM.

Compared to traditional methods, such as the Bayesian Information Criterion (BIC), the proposed k-value selection algorithm offers notable advantages in trajectory learning. The BIC is a popular method for model selection that balances model complexity with goodness of fit by penalizing models with more parameters, but it may not always perform well when the underlying clustering structure is ambiguous or difficult to discern. In contrast, our method addresses these challenges by enhancing the k-value selection process, improving both its robustness and accuracy in determining the optimal number of clusters.

Additionally, compared to deep learning approaches, which often require large datasets and extensive computational resources, the proposed method offers clear advantages. Deep learning techniques typically need vast amounts of labeled data to train robust models, whereas our method is based on unsupervised learning and can achieve high accuracy with relatively smaller datasets. This makes the GMM-KVS method particularly suitable for robotic tasks where data is limited or expensive to acquire.

## 3.2. Expectation maximization algorithm and Gaussian Mixture Regression

The Expectation Maximization algorithm is a method used to find the maximum likelihood estimates of parameters in probabilistic models with hidden variables [27]. The basic idea is to iteratively perform the Expectation step (E-step) and the Maximization step (M-step) to gradually increase the log-likelihood of the parameters. The specific contents of the E-step and M-step are as follows:

1) E-step: Calculate the posterior probability that each data point $\mathbf{x}_i$ belongs to the j-th Gaussian distribution:

$$\gamma_{jk}^{(t)} = \frac{\pi_k^{(t)}\phi(\mathbf{x}_j|\mu_k^{(t)},\Sigma_k^{(t)})}{\sum_{k=1}^{K}\pi_k^{(t)}\phi(\mathbf{x}_j|\mu_k^{(t)},\Sigma_k^{(t)})} \tag{8}$$

2) M-step: Re-estimate the parameters of the GMM using the posterior probabilities:

$$\pi_k^{(t+1)} = \frac{\sum_{j=1}^{n}\gamma_{jk}^{(t)}}{N} \tag{9}$$

$$\mu_k^{(t+1)} = \frac{\sum_{j=1}^{n}\gamma_{jk}^{(t)}\mathbf{x}_j}{\sum_{j=1}^{n}\gamma_{jk}^{(t)}} \tag{10}$$

$$\Sigma_k^{(t+1)} = \frac{\sum_{j=1}^{n}\gamma_{jk}^{(t)}(\mathbf{x}_j-\mu_k^{(t+1)})(\mathbf{x}_j-\mu_k^{(t+1)})^{\mathrm{T}}}{\sum_{j=1}^{n}\gamma_{jk}^{(t)}} \tag{11}$$

Repeat the E-step and M-step until the parameters converge or the predetermined number of iterations is reached to obtain the trained GMM parameters. Gaussian Mixture Regression (GMR) is a regression technique based on GMM, used for predicting outputs under given input conditions. Based on this, the corresponding predicted trajectories can be obtained for robotic arm movement according to the requirements of different trajectory planning tasks.

## 3.3. Method process

The proposed method, GMM-KVS, integrates Gaussian Mixture Models with a k-value selection algorithm for robotic arm trajectory learning. By learning from the original trajectory data, it can generate corresponding planned trajectories based on specific task requirements. Algorithm 1 outlines the detailed process of this method.

---

**Algorithm 1** The process of the proposed method

1: Obtain the original trajectory data of the robotic arm

2: # Optimal k-value selection process

3: **for** k = 1 to $K_m$ **do**

4:     Perform k-means clustering with the current k-value

5:     **for** i = 1 to k **do**

6:         $SSE_i = \sum_{p \in A_i}^{n} |p - a_i|^2$

7:     **end for**

8:     $S_{MP} = \sum_{i=1}^{k} e^{\frac{SSE_i}{\theta}} + e^{\frac{\max_{1 \le i \le k} SSE_i}{\theta}}$

9: **end for**

10: Plot a line chart with k and $S_{MP}$ , calculate the slopes of different segments, and select the k-value corresponding to the maximum difference between adjacent slopes as the optimal k-value $K_{op}$

11: # Initialization and iterative training of GMM parameters process

12: Perform k-means clustering with $K_{op}$ to obtain the initial parameters of the GMM

13: **while** not converged **do**

14:     **for** j = 1 to n **do**

15:         **for** k = 1 to $K_{op}$ **do**

16:             $\gamma_{jk} = \dfrac{\pi_k \phi(x_j | \mu_k, \Sigma_k)}{\sum_{k=1}^{K_{op}} \pi_k \phi(x_j | \mu_k, \Sigma_k)}$

17:         **end for**

18:     **end for**

19:     $N = \sum_{j=1}^{n} \sum_{k=1}^{K_{op}} \gamma_{jk}$

20:     **for** k = 1 to $K_{op}$ **do**

21:         $\pi_k = \dfrac{\sum_{j=1}^{n} \gamma_{jk}}{N}$

22:         $\mu_k = \dfrac{\sum_{j=1}^{n} \gamma_{jk} x_j}{\sum_{j=1}^{n} \gamma_{jk}}$

23:         $\sum_K = \dfrac{\sum_{j=1}^{n} \gamma_{jk} (x_j - \mu_k)(x_j - \mu_k)^T}{\sum_{j=1}^{n} \gamma_{jk}}$

24:     **end for**

25: **end while**

26: # Trajectory generation process

27: Obtain the requirements for the trajectory planning task

28: Generate the trajectory using GMR combined with the trained GMM parameters

---

# 4. Experimental results

## 4.1. Simulation experiment

The simulation experiment section uses three 2D theoretical nonlinear dynamic systems [28] to generate theoretical datasets, as follows:

a)  System 1

$$\begin{cases} \dot{X}_1 = -X_1 + 2X_1^2 X_2 \\ \dot{X}_2 = -X_2 \end{cases} \qquad (12)$$

b)  System 2

$$\begin{cases} \dot{X}_1 = -X_2 \\ \dot{X}_2 = X_1 - X_1^3 - X_2 \end{cases} \qquad (13)$$

c)  System 3

$$\begin{cases} \dot{X}_1 = -X_1 \\ \dot{X}_2 = -X_1 + (X_2 - \pi)\sin X_2 \end{cases} \qquad (14)$$

The theoretical demonstration trajectories of these three systems are shown in Fig 4. The trajectory points shown in the figure are selected for learning. Each system has an asymptotically stable point located at the origin within the data collection area.

The elbow method selects the k-value based on the SSE. According to (3), the SSE for each k-value within a given range is calculated for the three systems, and a line graph is plotted. The results are shown in Fig 5.

The k-value selection algorithm used in this paper modifies the SSE calculation formula. According to (7), the $S_{MP}$ for each k-value within a given range is calculated for the three systems, and a line graph is plotted. The results are shown in Fig 6.

As shown in Figs 5 and 6, when using the elbow method for k-value selection, it is not possible to intuitively determine the optimal k-value from the graphs for Systems 1-3, as there is no clear "elbow point" in the figures. However, by applying the k-value selection algorithm proposed in this paper, the limitations of the elbow method can be overcome. With

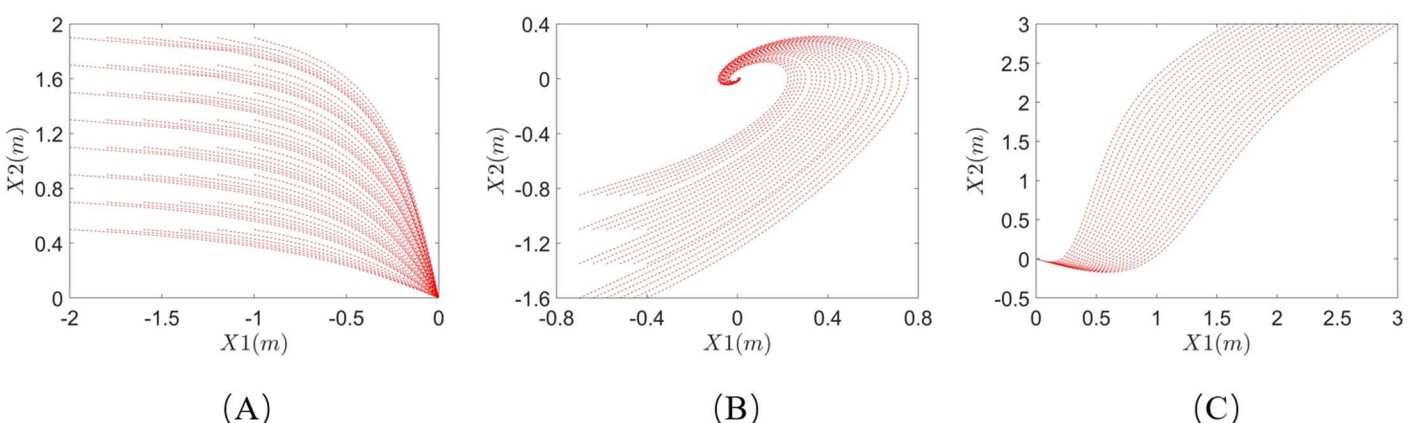

**Fig 4. Theoretical demonstration trajectories: (A) system 1; (B) system 2; (C) system 3.**

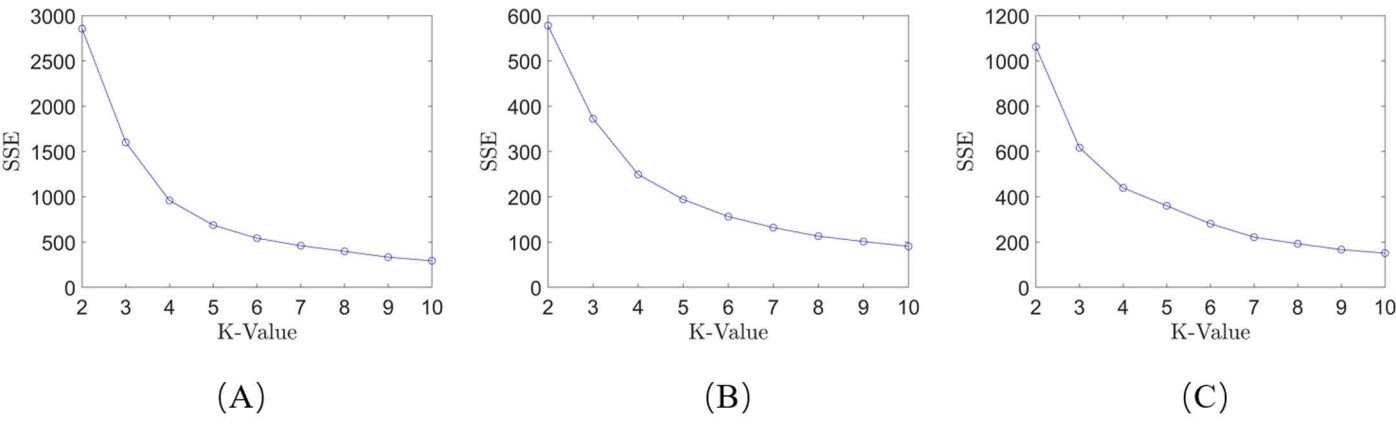

**Fig 5. Line chart plotted using the elbow method: (A) system 1; (B) system 2; (C) system 3.**

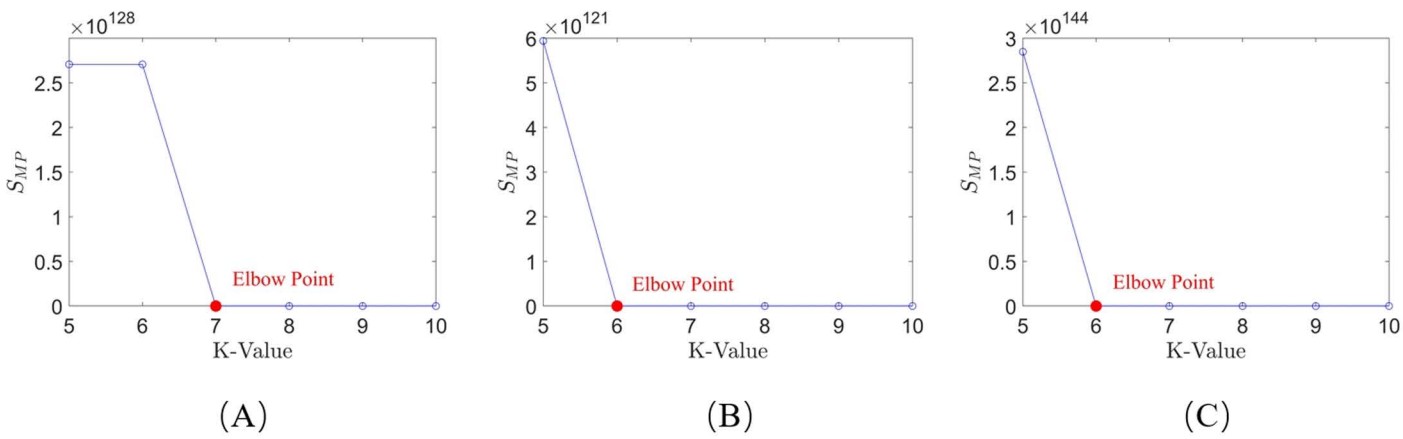

**Fig 6. Line chart plotted using the k-value selection algorithm: (A) system 1; (B) system 2; (C) system 3.**

this algorithm, the optimal k-value can be intuitively identified from the line graph, where the "elbow point" is clearly marked (as indicated by the red points). The optimal k-values for Systems 1, 2, and 3 are 7, 6, and 6, respectively. This demonstrates that the k-value selection algorithm used in this paper has a broader applicability compared to the elbow method and is significant for the final trajectory learning results.

As previously mentioned, for the three theoretical datasets selected in this paper, the k-value selection algorithm can be used to determine the optimal k-value. To further verify the effectiveness of the GMM-KVS method, this section compares it with the GMM method without the k-value selection algorithm and the Dynamic Movement Primitives (DMP) method [29]. For Systems 1-3, three starting points within the original trajectory collection range and three starting points outside the collection range (to verify generalization ability) are selected. Different trajectory imitation learning methods are used to generate trajectories from the starting points to the origin, and these trajectories are compared with the theoretical trajectories.

Figs 7, 8, and 9 illustrate the trajectory comparisons for system 1, system 2, and system 3, respectively. As shown in Figs 7–9, the GMM-KVS method can effectively complete trajectory

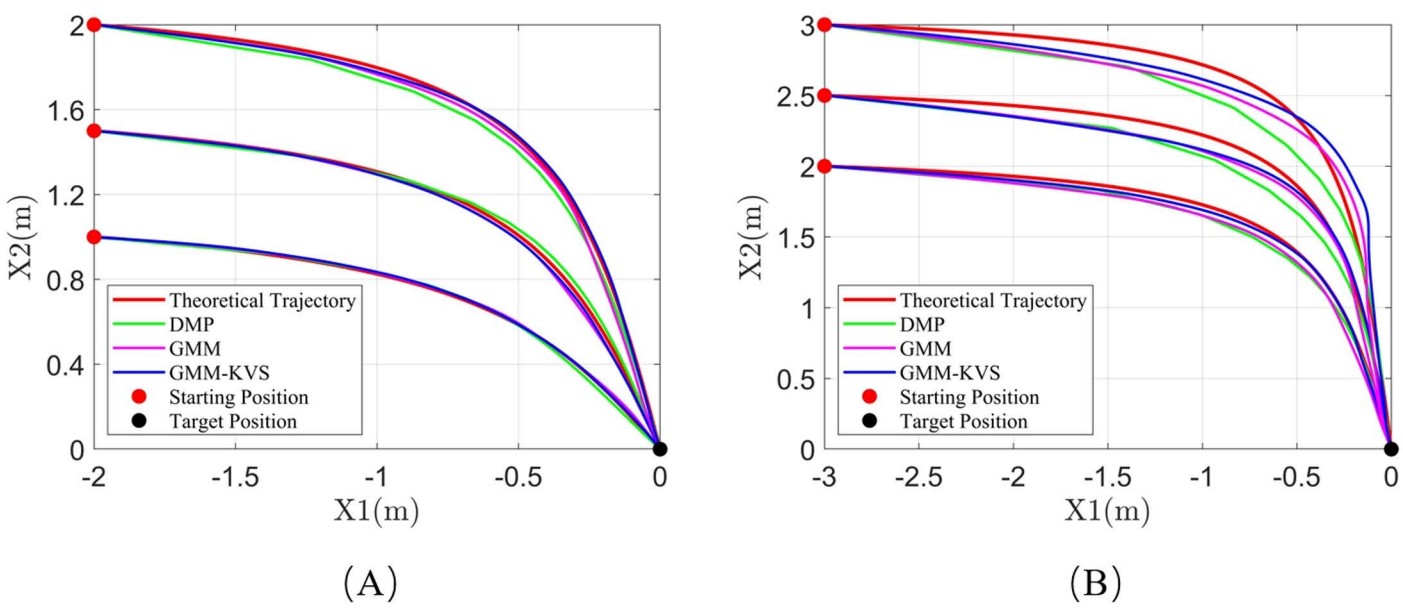

**Fig 7. Trajectory comparison for system 1: (A) starting point within collection range; (B) starting point outside collection range.**

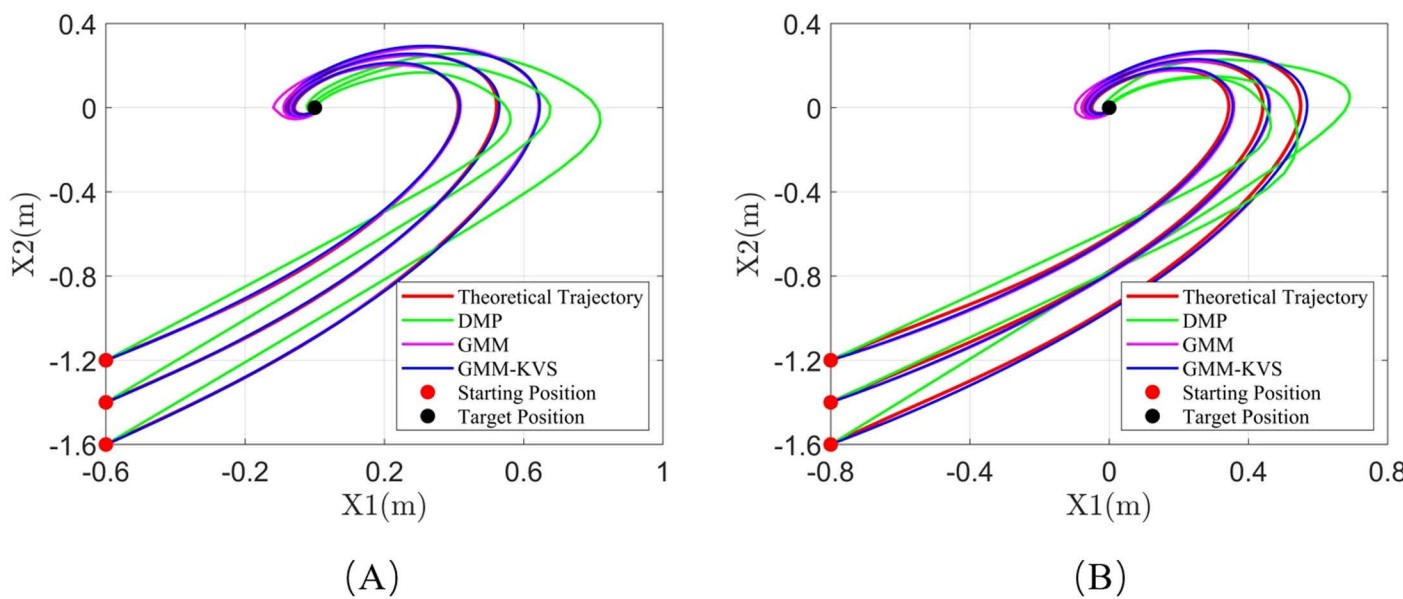

**Fig 8. Trajectory comparison for system 2: (A) starting point within collection range; (B) starting point outside collection range.**

prediction for both starting points within the original trajectory collection range and generalized starting points outside the collection range. The generated trajectories are superior to those produced by the GMM method and the DMP method.

To quantitatively analyze the superiority of the GMM-KVS method, trajectory errors are described using the Mean Absolute Error (MAE) [30] and Root Mean Square Error (RMSE) [31]. The calculation formulas are as follows:

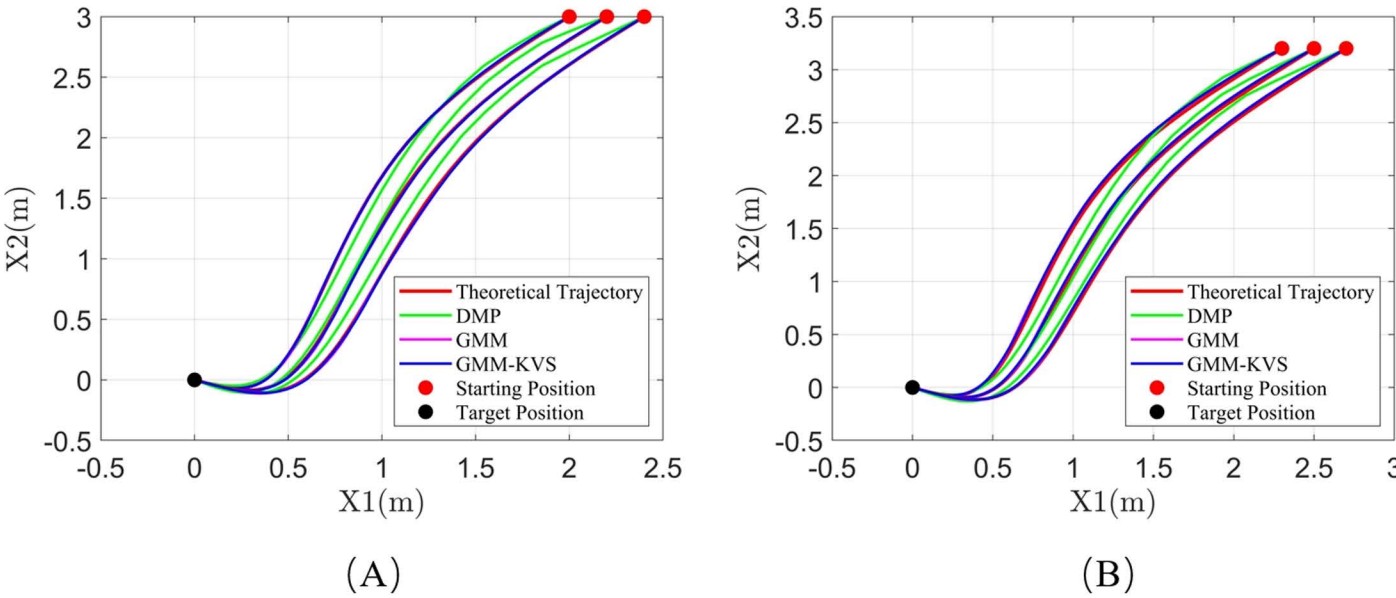

**Fig 9. Trajectory comparison for system 3: (A) starting point within collection range; (B) starting point outside collection range.**

$$MAE = \frac{1}{N}\sum_{i=1}^{N}\left|y\left(i\right) - \hat{y}\left(i\right)\right| \tag{15}$$

$$RMSE = \sqrt{\frac{1}{N}\sum_{i=1}^{N}\left(y\left(i\right) - \hat{y}\left(i\right)\right)^2} \tag{16}$$

where $y\left(i\right)$ and $\hat{y}\left(i\right)$ represent the predicted trajectory and the theoretical trajectory, respectively, and N represents the number of trajectory points. MAE directly reflects the average difference between the predicted trajectory and the theoretical trajectory, while RMSE emphasizes the impact of larger errors. Each starting point is numbered from 1 to 18, and the errors between the GMM-KVS, GMM, and DMP methods and the theoretical trajectory are calculated. The results are shown in Fig 10.

For each trajectory learning method, the arithmetic mean of MAE and RMSE is calculated, and the average of these two values is used as the overall evaluation value. The specific results are shown in the schedule S1 Table and S2 Table in S1 File, and the results in Table 1 highlight the performance of the three trajectory learning methods. The GMM-KVS method demonstrates significant improvement in trajectory accuracy, with reduced error values indicating better alignment with the target trajectories.

To further quantify the improvements, we calculated the 95% confidence intervals for both MAE and RMSE, which offer a more robust statistical basis for the findings. For MAE, the confidence intervals for the DMP, GMM, and GMM-KVS methods are [0.0436, 0.0705], [0.0157, 0.0327], and [0.0097, 0.0221], respectively. For RMSE, the confidence intervals are [0.0515, 0.0843], [0.0174, 0.0374], and [0.0107, 0.0261], respectively. These results demonstrate that the GMM-KVS method not only reduces the errors but also shows greater consistency, as evidenced by narrower confidence intervals. This statistical analysis confirms

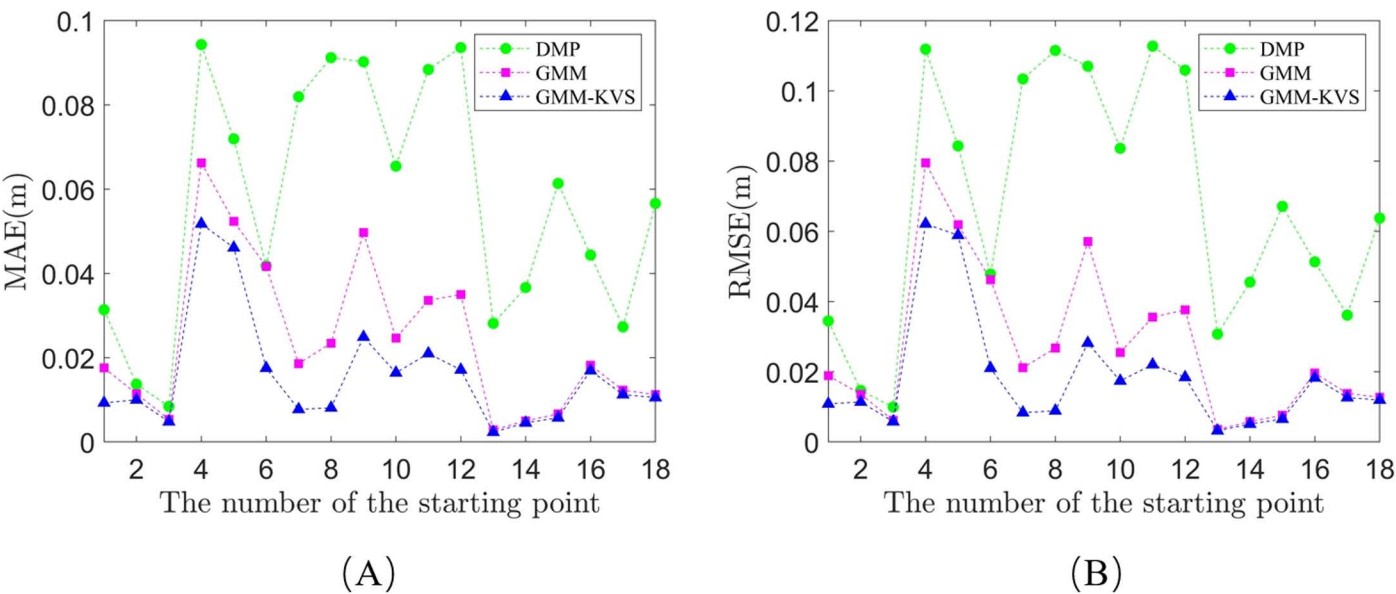

**Fig 10. Comparison of error values for trajectories generated by DMP, GMM, and GMM-KVS: (A) mean absolute error; (B) root mean square error.**

**Table 1. Evaluation values of DMP, GMM, and GMM-KVS (simulation experiment).**

| Method | Average MAE | Average RMSE | Overall evaluation value |
|---|---|---|---|
| DMP | 0.0570 | 0.0679 | 0.0625 |
| GMM | 0.0242 | 0.0274 | 0.0258 |
| GMM-KVS | 0.0159 | 0.0184 | 0.0171 |

the superior performance and reliability of the GMM-KVS method, further emphasizing its effectiveness for robotic arm trajectory learning.

## 4.2. Robotic arm experiment

To verify the effectiveness of the proposed method in real-world scenarios, experiments were conducted using a 6-DOF robotic arm, AUBO-C5. After picking up an object, the demonstrator drags the end-effector of the robotic arm to the specified position, as shown in Fig 11. The process was repeated 9 times, with each trial starting from a different initial position, but all trials aimed to reach the same target point. Although the trajectories were intended to be similar in shape, slight variations and small oscillations were inevitably observed due to factors such as minor human inconsistencies and the inherent flexibility of the robotic system. The trajectories were recorded and used as input for the imitation learning process, as shown in Fig 12.

Similar to simulation experiment, calculations and line charts were generated using the elbow method and the k-value selection algorithm proposed in this paper. The results are shown in Fig 13.

The results indicate that the k-value selection algorithm proposed in this paper performs better compared to the elbow method.

Similarly, to further validate the effectiveness of the GMM-KVS method, this section compares it with the GMM method that does not use a k-value selection algorithm and the

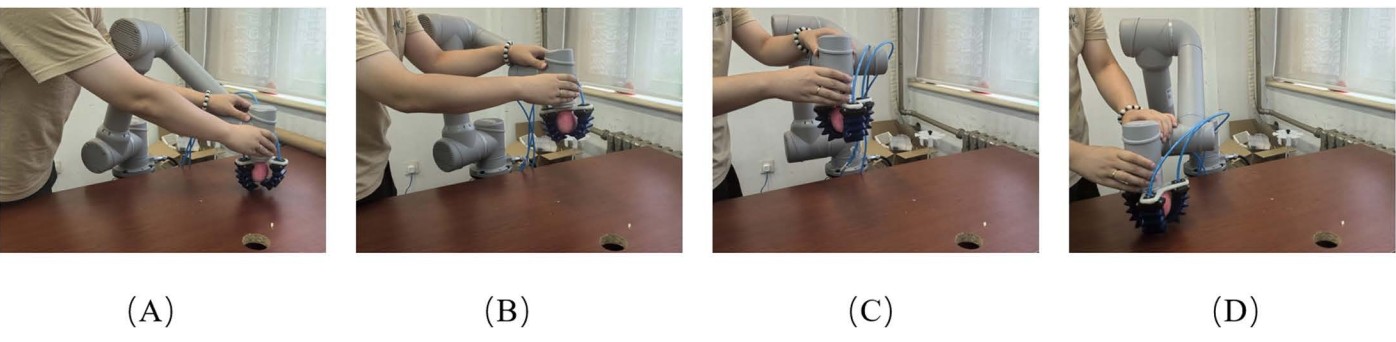

**Fig 11. Demonstration process of the robotic arm transporting an object: (A) starting position; (B) via position 1; (C) via position 2; (D) target position.**

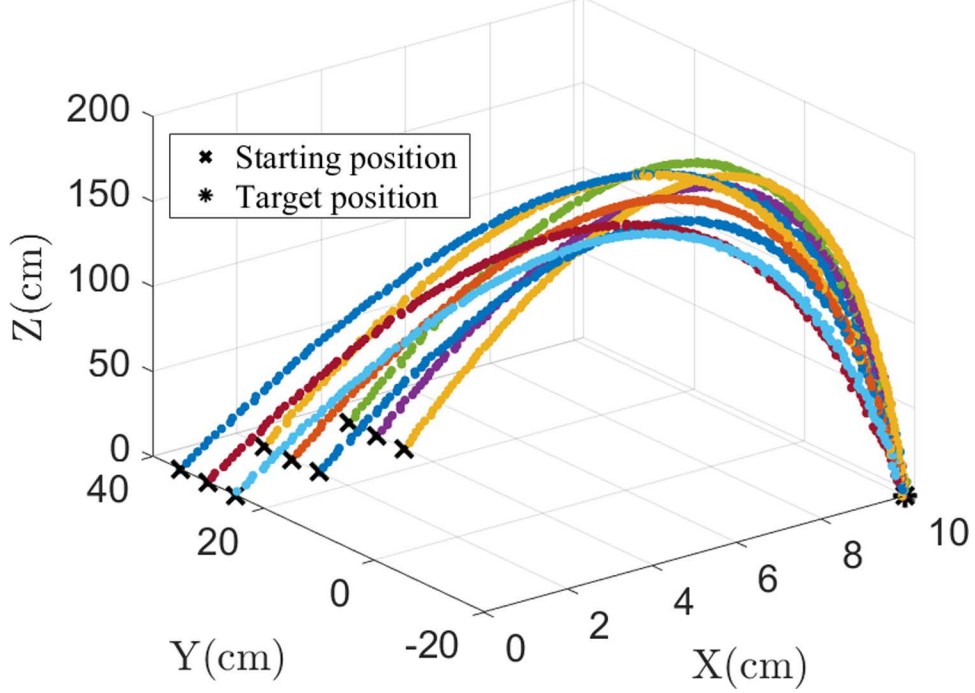

**Fig 12. Collected demonstration trajectories.**

DMP method. Trajectories were generated from three selected starting points of the collected trajectories and compared with the collected trajectories.

As shown in Fig 14, the trajectories generated using the GMM-KVS method exhibit smaller errors and are more similar to the original trajectories. To quantify the trajectory errors, the average MAE, average RMSE, and overall evaluation values for the three methods were calculated, as shown in Table 2, and the specific results are shown in the schedule S3 Table in S1 File. When comparing the results in Table 2 with those in Table 1, it is evident that the GMM-KVS method consistently outperforms both the GMM and DMP methods, showing significant improvements in both the simulation and robotic arm experiments. Notably, the GMM-KVS method demonstrates a greater improvement in the simulation experiment compared to the robotic arm experiment. This discrepancy can be attributed to the more

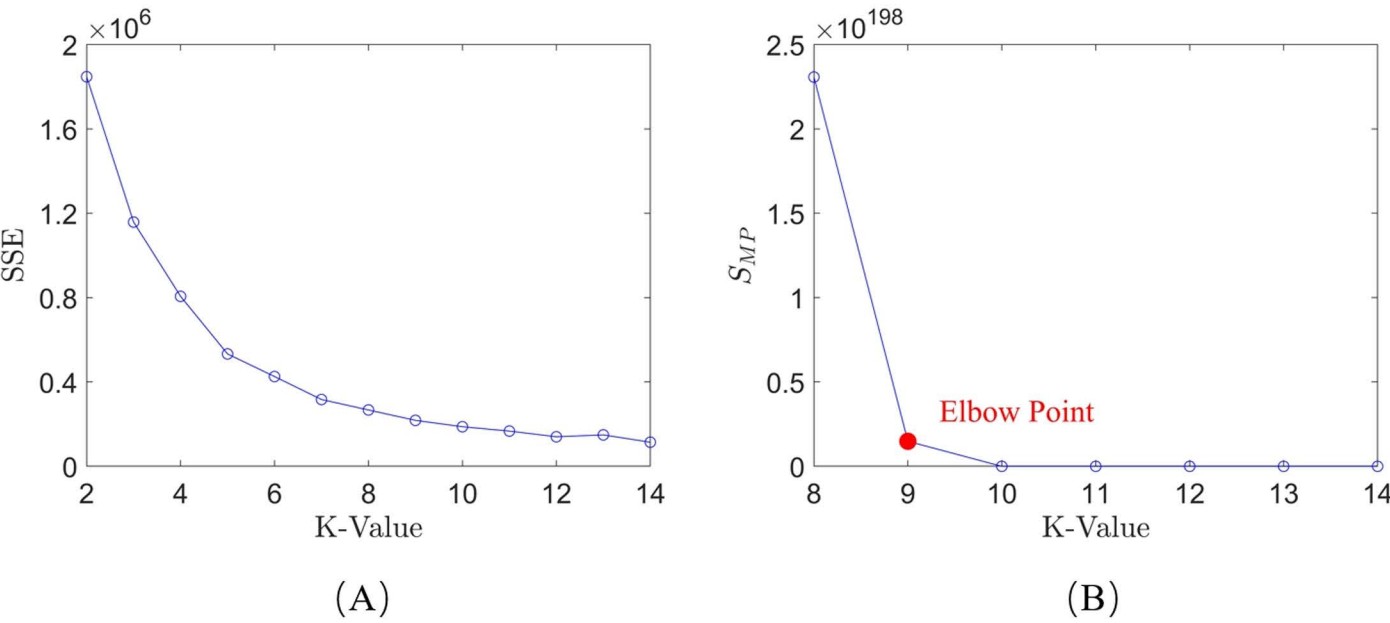

**Fig 13. Line charts of the elbow method and the k-value selection algorithm: (A) elbow method; (B) k-value selection algorithm.**

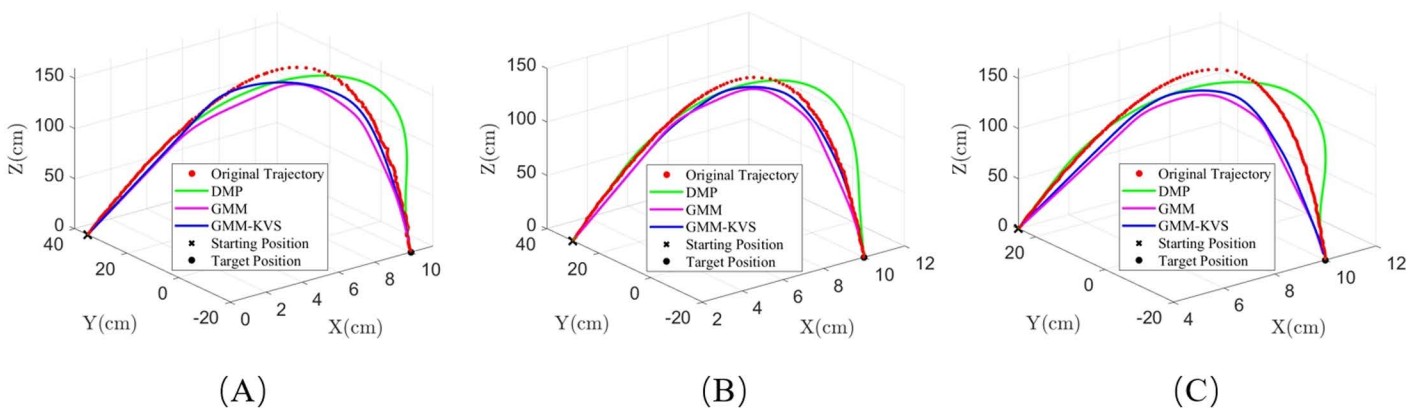

**Fig 14. Trajectory comparison chart: (A) starting point 1; (B) starting point 2; (C) starting point 3.**

**Table 2. Evaluation values of DMP, GMM, and GMM-KVS (robotic arm experiment).**

| Method | Average MAE | Average RMSE | Overall evaluation value |
|---|---|---|---|
| DMP | 8.4633 | 10.0753 | 9.2693 |
| GMM | 6.6747 | 7.7971 | 7.2359 |
| GMM-KVS | 5.3859 | 6.2729 | 5.8294 |

controlled and idealized conditions in the simulation, where factors such as noise and external disturbances are minimized. In contrast, the physical robotic arm experiment involves additional complexities, which may reduce the method's effectiveness. Despite these challenges, the GMM-KVS method still provides substantial improvements in the robotic arm experiment, highlighting its robustness in real-world applications.

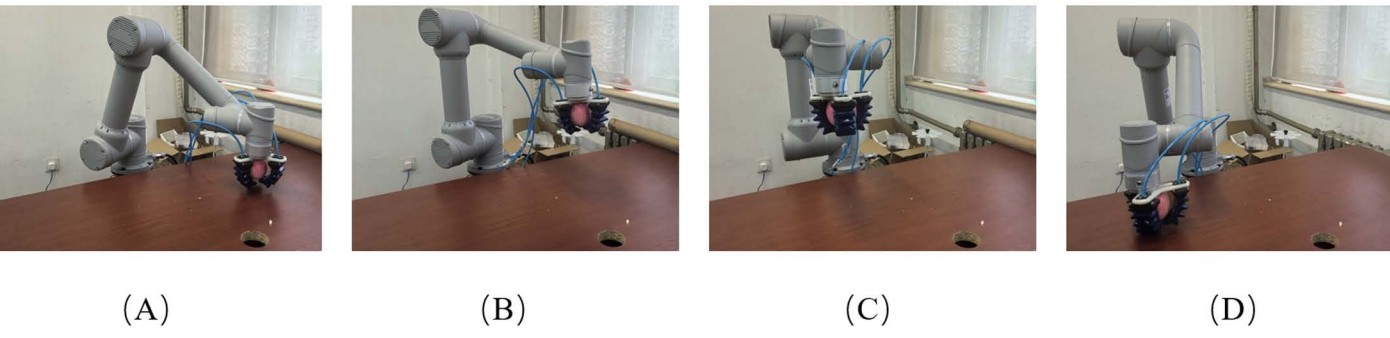

**Fig 15. The process of the robotic arm autonomously transporting an object: (A) starting position; (B) via position 1; (C) via position 2; (D) target position.**

By learning from the demonstrated trajectories, trajectories from different starting points to the target point can be generated, allowing the robotic arm to move along these trajectories, as illustrated in Fig 15.

Although the physical experiment in this study uses a specific 6-DOF robotic arm, the proposed method is primarily based on trajectory generation and is not limited to any particular robotic arm model. Given its focus on trajectory learning and optimization, the method can be applied to a wide range of robotic systems, including mobile robots and drones, demonstrating its broad applicability across different robotic platforms.

## 5. Conclusions

This paper presents a novel robotic arm trajectory learning method based on Gaussian Mixture Models combined with a k-value selection algorithm. By optimizing the initialization process through precise determination of the number of Gaussian kernels, the method avoids selection biases and significantly improves trajectory learning performance. Experimental results demonstrate that the proposed method enhances trajectory accuracy by over 15%, offering substantial improvements in precision and reliability compared to traditional approaches. These advancements contribute to the operational capability of robotic arms, particularly in complex and dynamic environments. The proposed method is not limited to a specific type or model of robotic arm, making it highly adaptable and suitable for a wide range of applications. It can be applied to various robotic systems, such as industrial assembly lines and autonomous logistics robots, where high precision and adaptability are essential for efficient operation across diverse tasks and environments.

While the proposed method shows significant improvements in trajectory accuracy, its performance depends on the quality of the initial trajectory data. The method may face challenges with noisy or incomplete data, and its computational complexity could be an issue when applied to larger datasets or real-time systems. Future work will focus on improving robustness to noise, enhancing scalability, and expanding the method's adaptability to a broader range of task scenarios and robotic systems. Additionally, efforts will be made to optimize the computational efficiency of the method to make it more suitable for real-time applications and large-scale datasets.

## Supporting information

**S1 File. Detailed results of MAE and RMSE for DMP, GMM, and GMM-KVS.**
(DOCX)

## Acknowledgments

The authors would like to thank the School of Technology, Beijing Forestry University for providing the research space.

## Author contributions

**Conceptualization:** Jingnan Yan, Yue Wu.

**Data curation:** Jingnan Yan.

**Funding acquisition:** Yue Wu.

**Methodology:** Jingnan Yan, Cheng Cheng.

**Project administration:** Yue Wu, Yili Zheng.

**Resources:** Yue Wu, Yili Zheng.

**Software:** Jingnan Yan.

**Validation:** Jingnan Yan, Kexin Ji.

**Writing – original draft:** Jingnan Yan.

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
