## [Decision Letter · Decision Letter 0]

4 Nov 2024

PONE-D-24-43885A novel trajectory learning method for robotic arms based on Gaussian Mixture Model and k-value selection algorithmPLOS ONE

Dear Dr. Wu,

Thank you for submitting your manuscript to PLOS ONE. After careful consideration, we feel that it has merit but does not fully meet PLOS ONE’s publication criteria as it currently stands. Therefore, we invite you to submit a revised version of the manuscript that addresses the points raised during the review process. According to the feedback from the reviewers, there are still several problems that need to be improved:1) The abstract and conclusion need further refinement. Meanwhile, a discussion on the application and limitation is suggested.2) The novelty of this manuscript should be clearly clarified.3) Most of the figures in this manuscript is not clear and a further adjustment is expected.4) More details about the methodology and experimental setup are missing.

We look forward to receiving your revised manuscript.

Kind regards,

Longhui Qin, Ph.D.

Academic Editor

PLOS ONE

When submitting your revision, we need you to address these additional requirements. 1. Please ensure that your manuscript meets PLOS ONE's style requirements, including those for file naming. The PLOS ONE style templates can be found at https://journals.plos.org/plosone/s/file?id=wjVg/PLOSOne_formatting_sample_main_body.pdf and https://journals.plos.org/plosone/s/file?id=ba62/PLOSOne_formatting_sample_title_authors_affiliations.pdf 2. Please include a caption for figure 1, Fig 2, Fig 3, Fig 4, Fig 5, Fig 6, Figs 7, Figs 8, Figs 9, Fig 10, Fig 11, Fig 12, Fig 13, Fig 14 and Fig 15. 3. Please note that PLOS ONE has specific guidelines on code sharing for submissions in which author-generated code underpins the findings in the manuscript. In these cases, we expect all author-generated code to be made available without restrictions upon publication of the work. Please review our guidelines at https://journals.plos.org/plosone/s/materials-and-software-sharing#loc-sharing-code and ensure that your code is shared in a way that follows best practice and facilitates reproducibility and reuse. 4. Thank you for stating the following financial disclosure:  [This work was supported by the National Natural Science Foundation of China under Grant. 62273053, Fundamental Research Funds for the Central Universities of China (BLX202128)].  Please state what role the funders took in the study.  If the funders had no role, please state: ""The funders had no role in study design, data collection and analysis, decision to publish, or preparation of the manuscript."" If this statement is not correct you must amend it as needed. Please include this amended Role of Funder statement in your cover letter; we will change the online submission form on your behalf. 5. We note that your Data Availability Statement is currently as follows: [All relevant data are within the manuscript and its Supporting Information files.] Please confirm at this time whether or not your submission contains all raw data required to replicate the results of your study. Authors must share the “minimal data set” for their submission. PLOS defines the minimal data set to consist of the data required to replicate all study findings reported in the article, as well as related metadata and methods (https://journals.plos.org/plosone/s/data-availability#loc-minimal-data-set-definition). For example, authors should submit the following data: - The values behind the means, standard deviations and other measures reported;- The values used to build graphs;- The points extracted from images for analysis. Authors do not need to submit their entire data set if only a portion of the data was used in the reported study. If your submission does not contain these data, please either upload them as Supporting Information files or deposit them to a stable, public repository and provide us with the relevant URLs, DOIs, or accession numbers. For a list of recommended repositories, please see https://journals.plos.org/plosone/s/recommended-repositories. If there are ethical or legal restrictions on sharing a de-identified data set, please explain them in detail (e.g., data contain potentially sensitive information, data are owned by a third-party organization, etc.) and who has imposed them (e.g., an ethics committee). Please also provide contact information for a data access committee, ethics committee, or other institutional body to which data requests may be sent. If data are owned by a third party, please indicate how others may request data access.

Reviewers' comments:

Reviewer's Responses to Questions

**Comments to the Author**

1. Is the manuscript technically sound, and do the data support the conclusions?

Reviewer #1: Yes

Reviewer #2: Yes

2. Has the statistical analysis been performed appropriately and rigorously? 

Reviewer #1: Yes

Reviewer #2: Yes

3. Have the authors made all data underlying the findings in their manuscript fully available?

Reviewer #1: Yes

Reviewer #2: No

4. Is the manuscript presented in an intelligible fashion and written in standard English?

Reviewer #1: Yes

Reviewer #2: Yes

5. Review Comments to the Author

Reviewer #1: The abstract need to be rewritten in more professional way, mention the gap of the science and the importance and the logic of this study, the methodology more detailed and the most important results such as quantities or percentage of improvement.

The conclusion needs to cover the most significant achievements, including some comparison in percentage or another professional way.

Considering to compare tables 1 and 2 data would provide some more useful information.

Consider reproduce the unclear pictures. Please revise all figures and make sure texts also have a good resolution.

Reviewer #2: The manuscript presents a novel approach for robotic arm trajectory learning using a Gaussian Mixture Model (GMM) with a k-value selection algorithm. This innovative combination addresses key challenges in trajectory accuracy. The research is technically sound and well-supported by experiments, but several improvements are suggested:

1. Clarify the Novelty: Expand on the unique aspects of the k-value selection method compared to existing approaches.

2. Methodology Details: Add further explanation for the mathematical steps and consider comparisons with other k-selection methods.

3. Experimental Clarity: Provide more details on the physical experiment setup and improve figure labeling.

4. Applications and Limitations: Discuss broader applications and potential limitations of the method.

5. Statistical Validation: Include confidence intervals to enhance the statistical robustness of the findings.

With these minor adjustments, the manuscript would be ready for publication in PLOS ONE, offering a valuable contribution to robotic trajectory learning.

6. PLOS authors have the option to publish the peer review history of their article (what does this mean? ). If published, this will include your full peer review and any attached files.

**Do you want your identity to be public for this peer review?** For information about this choice, including consent withdrawal, please see our Privacy Policy .

Reviewer #1: No

Reviewer #2: No

---

## [Author Response · Author response to Decision Letter 1]

2 Jan 2025

Thank you for your constructive feedback. We have completed the revisions, and the changes in the revised manuscript are highlighted in blue text. We have also addressed the comments from both the reviewers and the editorial team.

Response: 1) We have revised the abstract and conclusion to clarify the significance of our findings, including the scientific gap, the novelty of our method, and key results such as the 15% improvement in trajectory accuracy. We also added a discussion of the practical applications of our method and its limitations.

2) The novelty of our work is now more clearly explained. Specifically, we highlight the integration of the improved k-value selection algorithm with the Gaussian Mixture Model, which significantly enhances trajectory learning accuracy compared to traditional methods.

3) We have enhanced the resolution of all figures and ensured that the text within the figures is clear and legible, meeting the publication standards.

4) We have expanded the methodology and experimental setup sections to provide more comprehensive details on the GMM-KVS methods, as well as the experimental environment and data collection process.

We have completed the revision of the manuscript. In addition, we have responded to the comments of two reviewers in the email and the peer review feedback in the accompanying documents, and uploaded the responses to the system.

---

## [Editor Report · Decision Letter 1]

5 Jan 2025

PONE-D-24-43885R1A novel trajectory learning method for robotic arms based on Gaussian Mixture Model and k-value selection algorithmPLOS ONE

Dear Dr. Wu,

Thank you for submitting your manuscript to PLOS ONE. After careful consideration, we feel that it has merit but does not fully meet PLOS ONE’s publication criteria as it currently stands. Therefore, we invite you to submit a revised version of the manuscript that addresses the points raised during the review process.

 Most of the reviewers' comments have been addressed. However, quality of all the figures seems not enough. It is suggested to improve them further, e.g., higher resolution. The font size in figures should be similar to that in the context. 

We look forward to receiving your revised manuscript.

Kind regards,

Longhui Qin, Ph.D.

Academic Editor

PLOS ONE
---

## [Author Response · Author response to Decision Letter 2]

14 Jan 2025

Thank you for your constructive feedback. We have enhanced the resolution of all figures to ensure clarity and quality. Additionally, the font sizes in all figures have been adjusted for consistency and improved readability. The updated figures were verified using the Preflight Analysis and Conversion Engine (PACE) tool to meet publication standards.

---

## [Editor Report · Decision Letter 2]

16 Jan 2025

A novel trajectory learning method for robotic arms based on Gaussian Mixture Model and k-value selection algorithm

PONE-D-24-43885R2

Dear Dr. Wu,

We’re pleased to inform you that your manuscript has been judged scientifically suitable for publication and will be formally accepted for publication once it meets all outstanding technical requirements.

Kind regards,

Longhui Qin, Ph.D.

Academic Editor

PLOS ONE
---

## [Editor Report · Acceptance letter]

PONE-D-24-43885R2

PLOS ONE

Dear Dr. Wu,

I'm pleased to inform you that your manuscript has been deemed suitable for publication in PLOS ONE. Congratulations! Your manuscript is now being handed over to our production team.

Kind regards,

on behalf of

Prof. Longhui Qin

Academic Editor

PLOS ONE